# Psychological Distress and Psychosocial Factors in the Non-Formal Context of Basketball Coaches in Times of the COVID-19 Pandemic

**DOI:** 10.3390/ijerph18168722

**Published:** 2021-08-18

**Authors:** César Torres-Martín, Inmaculada Alemany-Arrebola, Manuel Enrique Lorenzo-Martín, Ángel Custodio Mingorance-Estrada

**Affiliations:** 1Department of Didactics and School Organization, Faculty of Education Sciences, University of Granada, 18071 Granada, Spain; cesartm@ugr.es; 2Department of Developmental Psychology and Education, Faculty of Education and Sports Sciences, University of Granada, 52005 Melilla, Spain; 3Company of Mary School, 18014 Granada, Spain; profesor.manuel.lorenzo@gmail.com; 4Department of Didactics and School Organization, Faculty of Education and Sports Sciences, University of Granada, 52005 Melilla, Spain; amingoe@ugr.es

**Keywords:** coaches, non-formal education, health, psychosocial factors, stress

## Abstract

Psychological distress and psychosocial factors are studied in the sports context in players of various specialties, but are only little studied with coaches who carry out their work with these athletes; that is where we put the emphasis, trying to determine the perception of coaches on psychological distress and psychosocial risk factors that may influence their sports work in times of a pandemic. It is an ex post facto study with a single-group retrospective design, with a representative sample of 94 coaches out of a possible 109. The Kessler Psychological Distress Scale and the short version of the ISTAS21 Psychosocial Risk Assessment at Work Questionnaire were adapted to the sports context. The results show that the youngest, those with the least experience and level 1 and level 2 coaches show the highest levels of stress. According to the psychosocial risk assessment, level 1 and 2 coaches, with experience between 6 to 10 years, are in the risk zone. Therefore, it is important to work with a group of coaches who are in the psychosocial risk zone and with high levels of psychological discomfort in order to avoid mental, emotional and physical stress, for the good performance of their work in the best possible conditions.

## 1. Introduction

The first works on stress were published in the 1930s [1,2], and since then, there has been a great deal of research that justifies its relationship with psychosocial factors and health disorders, which are increased when trying to achieve improvements in performance, being a multifactorial concept that must be understood globally and encompass all athletes (coaches, players, athletes…), and not only players and athletes, as it encompasses factors such as knowledge and physical, tactical, technical, psychological and social practice to provide effective responses to competitive situations.

In 1984, the International Labor Organization (ILO) and the World Health Organization (WHO), through a Mixed Committee, systematically analyzed the progress of studies on psychosocial factors at work, and concluded the perceptions and experiences of employees to that effect identified in the interactions of work and its environment, job satisfaction and conditions of the entity, as well as the qualities of the workers and their needs, culture and personal situations outside of work.

Currently, the nature of these studies in work contexts in general is a growing concern [3,4,5,6,7,8,9,10,11,12]. There has also been an increase in research and practice related to psychosocial and health problems in sports, in particular, in recent years [13,14,15,16,17,18,19], especially in Australia, Europe and the United States. However, when it comes to coaches, it is not easy to review research in this regard, references are scarce [20,21,22,23], hence the importance of this study.

Stress is a characteristic of sports. As well as training the physical and technical demands of competition, it is also necessary to train psychosocial stress [24]. The conditions to which the coach is exposed, i.e., the psychosocial work factors, are the reason for the production of stress, and if they take root and become permanent, they transform into the well-known burnout syndrome [21], which occurs more frequently in work activities with direct contact between people [25].

Psychosocial factors are related to the development of a person’s well-being and health, being influenced by various internal and external conditions of the person, and so does the link between the coach and an athlete as sportspeople working together [26] as it is inevitable to establish relationships. Such conditions of the person are associated with the relationships between the work environment (work organization, activity process and satisfaction associated with the task) and the worker (competencies, limitations and private life) [27].

The coach may appreciate his/her functions as problematic when he/she is concerned about the development of his/her athletes and the achievement of his/her objectives so he/she cannot effectively address the inconveniences that occur in the daily performance of his/her work [28,29]. Consequently, he/she must develop the skills and competencies inherent to interrelationships so that personal and professional interaction is profitable when resolving possible conflicts or stressful situations [30]. In this personal relationship, the athlete seeks to learn technical–tactical aspects, be competent, be successful and achieve satisfaction, and the coach seeks to transmit knowledge and experiences, improve the athlete, and also success and satisfaction [22], being competent in the exercise of his/her professional development. In short, the coach and the athlete are mutually and causally interdependent, the behaviors, thoughts and feelings of one affect the behaviors, thoughts and feelings of the other [31].

At present, the pandemic caused by COVID-19 has undoubtedly been an additional psychosocial factor. The fear of new epidemic waves has been a complex problem that has affected society in general and the sports field in particular. New epidemics can cause considerable damage to countries, companies and individuals at different levels [32]. In this line, the sanitary emergency has deprived coaches and players of continuing with their usual routines, which can cause stress levels related to anxiety compared to those experienced on a daily basis [33,34], as well as other derived problems that have some similarities with those suffered by athletes during the different phases of a sports injury [35].

From the research found on human confinement, we can highlight the works carried out with prisoners in penitentiary centers and with the development of tasks in a similar situation of confinement, such as the studies at space stations with astronauts and with submarine crews. These studies show the consequences of isolation with symptoms related to low mood, anxiety and stress, fatigue and apathy, sleep problems, nightmares, migraines, fatigue and dizziness, palpitations, loss of appetite, as well as irrational anger, hypersensitivity to stimuli, confused thought process, hallucinations and suicidal thoughts, in addition to social isolation, difficulty adapting to an extreme context that differs from the usual environment [36,37,38,39,40]. Although there may be negative psychological consequences with confinement, there may also be protective factors, such as the resilience of individuals, that can minimize its effect and improve their well-being [41].

Considering the transmission characteristics and high contagiousness of COVID-19 (coughing, sneezing, air containment in enclosed spaces, personal contact, touching contaminated surfaces and then touching the eyes, nose or mouth, and touching items contaminated with fecal matter) [42,43,44], this type of person-to-person transmission has led to a strong distrust in interpersonal relationships, which has meant avoidance of contact [32] and has consequently affected interpersonal relationships.

The consequences of the pandemic, such as loss of financial stability and social connectivity, go beyond the immediate physical danger of the virus, transforming the way people live, work, socialize and cope with stressors in daily life [45]. The sum of pandemic-related stress and uncertainty has had a significant impact on mental health and well-being [46].

Psychosocial factors are not detected instantly; rather, this process takes time since by their nature they are perceived once their consequences are manifested, which means their prevention as a control strategy is not possible or is very difficult. In this sense, the models that account for this have also been expanding, from the so-called Demand–Control–Social Support model [47] to ISTAS21 [48,49,50], through the effort compensation imbalance model [51,52] and the COPSOQ [53].

Therefore, the pandemic syndrome suffered by coaches, players and athletes, associated with states of psychological distress and psychosocial factors, is considered a dynamic and complex process with biological, psychological and social effects that disrupt people’s health and produce psychosomatic alterations and psychological disorders in sportspeople [54].

There is research that analyzes the psychosocial problems affecting athletes [35,55,56,57] but few or nonexistent research that analyzes the levels of stress suffered by coaches. For this reason, this research focuses on determining the level of psychological risks and analyzing the psychosocial risks of basketball coaches in their professional development during the current 2020–2021 season characterized by the COVID-19 pandemic conditions.

## 2. Materials and Methods

### 2.1. Design

The methodology used is empirical–analytical, using an ex post facto design for a single-group retrospective study [58]. Regarding the data collection process, a cross-sectional design was used.

### 2.2. Participants

A total of 94 basketball coaches from the province of Granada (Spain) participated in this study, with a total population of 109; therefore, the sampling error is 3.7%, with the confidence level of 95%. The sample under study is made up of 78 men (83%) and 16 women (17%). The mean age is 32.84 (SD = 11.26), ranging from 18 to 59 years.

The level of education varied, although the majority, 64 of those surveyed, completed university studies (68.08%), and 30 completed vocational training (middle and higher), high school and compulsory secondary education (31.91%).

With regard to basketball qualifications, 42 of the participants (44.7%) indicated that they are level 1 monitors, 34 are level 2 coaches (36.2%) and 18 are level 3 or top coaches (19.1%).

On the other hand, the years of experience of the respondents ranged from 1 to 33 years, with the mean experience being 9.27 years (SD = 7.88 years).

The following table details the participants according to their basketball qualification and years of experience (Table 1).

### 2.3. Instrument

For this research, we relied on the Kessler Psychological Distress Scale (k10) [59], the questions whereof describe ways in which one acts or feels according to the current situation of the coaches investigated, and on the short version of the Psychosocial Risk Assessment at Work ISTAS21 (COPSOQ) questionnaire designed to initiate risk assessment in small entities with less than 25 workers, in this case, in basketball sports clubs and their coaches, adapting both instruments to the sports environment as an example of non-formal education in the COVID-19 times.

The scale used is the Spanish adaptation of the Kessler’s scale [60] with the internal consistency index of α = 0.82, which measures the stress experienced by coaches. It consists of 10 items with four response options, from “never” to “always”, so that the higher the score, the greater the perceived stress, the minimum score being 10, the maximum—40.

The second instrument used is that of Kristensen [61], adapted to the non-formal contexts, specifically to the sports context, and consisting of six dimensions:-Cognitive and emotional psychosocial exigencies, i.e., it includes the amount of work, time pressure and attention level.-The coach’s perceived control over his/her work, as well as the possibilities of developing his/her personal skills.-Insecurity in terms of both his/her work and the recognition he/she gets himself/herself.-Social support and leadership quality, including both the support received from peers and superiors, as well as role clarity and role conflict.-The coach’s perception of the difficulty of balancing training and family.-Perception of fairness and trust.

The response options were from 1 to 4, with 1 being “never”, 4—always. The items were summed separately, giving a total score placing a person in one of the three zones: green (low risk), yellow (moderate risk) or red (high risk).

### 2.4. Procedure

In order to deliver the questionnaires, the coordinators of the sports clubs were contacted by means of a letter of introduction by email, to which they kindly responded. After this presentation and with the approval of the clubs, we proceeded to sending the online questionnaires to the monitors (level 1), coaches (level 2) and top coaches (level 3) from each of the clubs that participated voluntarily and freely through the suggestion of their coordinators.

For data analysis, SPSS version 23.0 (IBM, Chicago, IL, USA) was used to perform the descriptive, reliability and correlation statistics. In addition, the sample fit to the normal distribution was analyzed using the Kolmogorov–Smirnov test with the Shapiro–Wilk correction and homoscedasticity using Levene’s test. Since the data did not fit the normal curve, nonparametric statistical analyses of contrasts were performed (Mann–Whitney *U* and Kruskal-Wallis *H* tests). A 95% confidence interval was used to detect significance.

## 3. Results

First, the descriptive statistics of the Psychological Distress in Non-Formal Contexts (PD-NFC) questionnaire were analyzed and the data are shown in Table 2. The results show that the skewness was in all the cases positive, which shows that there was a higher concentration of responses denoting low scores on all items, i.e., “never” or “almost never”. In relation to reliability measured through the internal consistency index, Cronbach’s alpha is 0.878, with the corrected item-total correlation ranging from 0.340 to 0.734.

Regarding the sex variable, the data indicate that there were no significant differences. Regarding the age variable, the results show significant differences (Kruskal-Wallis *H* = 8.556; *p* < 0.001, with the group of coaches aged 18 to 23 years showing the highest stress indices ($\bar{x}$ = 16.81) and the lowest levels observed in those aged 41 to 50 years ($\bar{x}$ = 11.67), followed by the age group 51–60 years ($\bar{x}$ = 14.00) (Appendix A: Table A1, Figure A1 and Figure A2).

Regarding the basketball qualification variable, significant differences were observed (Kruskal-Wallis *H* = 16.875; *p* < 0.001), with the group of level 2 coaches ($\bar{x}$ = 16.26) showing the highest levels of stress, and the lowest observed among the top coaches or level 3 ($\bar{x}$ = 12.00) (Appendix A: Table A2, Figure A3 and Figure A4).

Analyzing the years of experience variable, the data indicate that the mean was 9.27 (SD = 7.88), ranging from one year (11.7%) to those indicating 33 years of experience (1.1%). The results show significant differences (Kruskal-Wallis *H* = 12.848; *p* < 0.005), with the group with more years of experience showing lower levels of stress ($\bar{x}$ = 32.10), and the highest observed among those with 1–5 years of experience ($\bar{x}$ = 54.77) (Appendix A: Table A3, Figure A5 and Figure A6).

On the PD-NFC scale, the participants were grouped into three grades: grade 1 corresponds to low psychological distress in coaches, i.e., the participants’ means were below the 25th percentile ($\bar{x}$ ≤ 12); grade 2, moderate level of psychological distress, the means are between the 26th and 74th percentiles ($\bar{x}$ ≥ 13 to $\bar{x}$ ≤ 16); finally, grade 3, coaches with high levels of psychological distress, is characterized by the means above the 75th percentile ($\bar{x}$ ≥ 17). The data indicate that there is an inversely proportional relationship between the level of stress, degree and experience (Table 3). Thus, coaches with level 1 and 2 qualifications show higher levels of stress than level 3 coaches, i.e., those with less basketball qualifications and fewer years of experience show the highest levels of stress before competition.

The results of the Psychosocial Risk Assessment Questionnaire in Non-Formal Contexts (PRAQ-NFC) indicate that the asymmetry was positive in dimensions 2, 4 and 6, which shows that there was a greater concentration of such responses as “many times” or “always”; on the contrary, in dimensions 1, 3 and 5, the response options were concentrated around “sometimes” or “never”. In terms of reliability measured using the internal consistency index, Cronbach’s alpha ranged from 0.422 to 0.853 (Table 4).

Subsequently, the sections of the PRAQ-NFC were analyzed according to its component dimensions. For this purpose, in each section, the scores were classified into three exactly equal groups following the criteria of Kristensen [61] (Table 5), giving rise to the three zones described: green (low psychosocial risk), yellow (moderate psychosocial risk) and red (high psychosocial risk).

On the other hand, in the inferential analyses, the data indicate that there were no significant differences according to the sex variable. On the contrary, in relation to the basketball qualification variable, there were differences in the sections “Psychosocial demands”, “Active work and development possibilities” and “Social support and leadership quality” (Appendix A: Table A4, Figure A7, Figure A8 and Figure A9). Regarding the section “Psychosocial demands” (Kruskal-Wallis *H* = 16.381; *p* < 0.00), the data indicate that it was the level 2 coaches ($\bar{x}$ = 12.41) who obtained the highest scores and, therefore, were in the range of the scores that fall in the red zone. Regarding “Active work and developmental possibilities” (Kruskal-Wallis *H* = 11.218; *p* < 0.005), the highest scores were obtained by level 3 coaches ($\bar{x}$ = 35.94), with the same tendency occurring in the dimension “Social support and leadership quality” (Kruskal-Wallis *H* = 9.826; *p* < 0.05; $\bar{x}$ = 33.05), this group of coaches being in the green zone.

In reference to the years of experience in the field of sports training, the data indicate significant differences in sections 1 “Psychosocial demands” (Kruskal-Wallis *H* = 6.127; *p* = 0.047) and 2 “Active work and development possibilities” (Kruskal-Wallis *H* = 7.339; *p* = 0.025). In the first section, it was the group with 6–10 years of experience that obtained the highest scores ($\bar{x}$ = 11.9), and the lowest scores were observed in the group with 1 to 5 years of experience ($\bar{x}$ = 10.3), the former being in the yellow zone and the latter in the green zone. Furthermore, regarding section 2 “Active work and development possibilities”, it was the participants with more than 11 years of experience who showed the highest scores ($\bar{x}$ = 34.06), and the lowest scores were found in the group with 6–10 years of experience ($\bar{x}$ = 30.95), both being in the yellow zone.

In addition, the correlation between stress levels and the sections of the PRAQ-NFC was analyzed, showing that there is a directly proportional relationship between “Psychosocial demands” (*r* = 0.265; *p* = 0.010) and “Insecurity” (*r* = 0.303; *p* = 0.003) and an inversely proportional relationship between “Active work and development possibilities” (*r* = −0.263; *p* = 0.011) and “Social support and leadership quality” (*r* = −0.238; *p* = 0.021).

Taking stress levels as an independent variable, the data reveal significant differences in all the sections of the PRAQ-NFC, except in 5, “Double presence” (Table 6), all the sections being in the yellow zone.

Finally, the profile of the participants with the highest scores on the PD-NFC scale, i.e., with the means above the 75th percentile (≥17), was analyzed. The data show that there were 24 participants (25.53%) who had a high level of stress, mostly men, aged between 18 and 23 years. In terms of qualification, level 2 coaches experienced greater stress, with a range of experience between 1 and 5 years for the most part. In addition, level 1 and 2 coaches with fewer years of experience had higher levels of precompetition stress.

Of the 24 coaches presenting high levels of stress, eight were level 1 monitors (33.3%) and 16 were level 2 coaches (66.6%). The profile of these two groups is analyzed below in terms of the scores obtained in the PRAQ-NFC.

In relation to level 1 monitors, in dimension 1 “Psychosocial demands”, it was mainly women who experienced high cognitive and emotional psychosocial demands. Thus, 87% indicated the large amount of work to be carried out along with the time pressure to perform it (62.5%), feeling pressured by both circumstances, which requires a high attention level. In addition, 75% reported taking problems home with them, with a significant emotional toll on 50% of the respondents, with 25% of them hiding their emotions.

In dimension 2 “Active work and possibilities for development”, the data indicate that 50% perceived having control over their work and possibilities for developing their personal skills; 75% showed initiative to develop their work, although it should be noted that 25% had difficulties being absent from trainings for one hour, having to ask for special permission.

In dimension 3 “Insecurity”, 25% perceived insecurity in their job and lack of recognition. Fifty percent of those surveyed were concerned about possible changes in their working conditions and only 12.5% expressed concern about the economic aspect.

In dimension 4 “Social support and leadership quality”, including both the support received from peers and superiors, as well as role clarity and conflict, 37.5% were concerned that they would not be able to continue working at the club and 50% were concerned about the change of work schedule and days.

In relation to dimension 5 “Double presence”, 37.5% indicated that they had difficulty balancing training sessions with the family. With regard to family and household chores, only 25% responded that they took care of most of them and 50% reported that if they missed home that day for some reason they were left undone. In addition, 62.5% considered that sometimes they needed to be at home and at the club at the same time.

In dimension 6 “Esteem”, 50%, mainly women, indicated the need for the perception of equal justice and trust. In addition, 12.5% considered that they had never received the support they needed in difficult situations.

Regarding the data collected in the group of level 2 coaches, in dimension 1 “Psychosocial demands”, 68.75% highlighted the amount of work that coaches had to perform and the time pressure during the same; 50.1% found it difficult to forget the problems that took place during training sessions, with 50% reporting emotional exhaustion and 43.8% responding that they hid their emotions in the vast majority of situations.

In dimension 2 “Active work and development possibilities”, 31.25% perceived having control over their work and that it helped them to develop their personal skills, although 18.8% felt that their opinion was not taken into account very much. In addition, 12.5% reported that they had to ask for special permission to resolve a personal or family matter during training hours.

In dimension 3 “Insecurity”, the data indicated that coaches perceived a great deal of uncertainty with their work. Thus, 31.3% were concerned about how difficult it would be to find another club if they could not continue with their current one, and 56.3% indicated that they were worried about having to perform tasks against their will. Furthermore, 31.3% were concerned about the changes that could occur due to the economic remuneration not being updated or having their salary lowered, and 62.6% of the coaches were concerned about schedule changes.

In dimension 4 “Social support and leadership quality”, the data show that coaches perceived a lack of social support and leadership. We found that 18.8% considered that only sometimes they received support from their superiors and only 12.5% indicated that sometimes they did not have help from their colleagues. As for the information they received, 31.3% responded that they only received it some of the time from their immediate superiors. On the other hand, 43.8% considered that their bosses did not plan their work well.

In dimension 5 “Double presence”, 43.75% showed difficulty in balancing their work with their family since 62.5% of the respondents carried out most of the domestic and family chores, and 37.5% thought about domestic chores during training sessions; 55.8% would often need to be at home and at the club at the same time.

In dimension 6 “Esteem”, 18.75% felt that they lacked the recognition they deserved, and only 6.3% of the coaches felt that the club treated them unfairly; 93% of the coaches felt that the club supported them in difficult times.

## 4. Discussion

The development and application of measurement instruments is a common task in socio-educational research, in its different fields of intervention, whether formal, non-formal or informal. For the present work, the adaptation of the Psychological Distress in Non-Formal Contexts scale (PD-NFC) was carried out, the reliability analysis showed a very adequate index [62]; consistency indices above 0.80 were considered very acceptable. Therefore, it can be concluded that the PD-NFC scale presents an adequate degree of reliability to measure the level of psychological distress of basketball coaches involved with different sports clubs in the non-formal setting in this pandemic time. In addition, it is necessary to develop instruments to apply in the non-formal sports context since there is little research that analyzes the realities of the people who work in the world of sports [21,22] despite them having a significant role in directing the processes of preparation and competition of athletes [63].

The data indicate that there were no differences between coaches with respect to the sex variable. On the contrary, age and years of experience were related to stress levels, being higher in younger coaches and in those with less experience, showing the highest levels of stress in their professional performance. For this reason, the levels of psychological distress in the COVID-19 time shown by coaches should be analyzed in order to prevent it leading to the burnout syndrome [21] and affecting negatively the performance of their work, with the pandemic being an aggravating factor that aggravates the situation of anxiety [33,34].

Regarding the qualification variable, level 2 coaches showed the highest levels of stress, and level 3 coaches had the lowest scores. A possible explanation is due to the considerable background in their practical knowledge, group management and social and sporting recognition, which can lead to success and satisfaction [22] that may be impaired by the effects of the pandemic. Therefore, the degree of experience as a coach involves addressing more facets of the athlete in addition to physical condition [64], which, as a consequence of confinement, may be altered, worrying about their psychological well-being which due to COVID-19 may encounter moments of uneasiness, anxiety and fear, that is, the coach helps the sportsman to improve as a person, which means less stress for the coach himself and more ability to control the situation, which is difficult to manage in the COVID-19 times.

On the other hand, we must highlight the existence of an inversely proportional relationship between motivation and the coaches’ years of experience. The most motivated were level 1 coaches, mainly due to the fact that the beginnings of a new journey are exciting and full of positive expectations. However, it should be noted that coaches with less experience, between 1 and 5 years, were those who showed the highest levels of stress, which can be explained by the uncertainty caused by the new training situation characterized by social distancing and the measures imposed by the COVID-19 pandemic, which makes this stage different in terms of training times and procedures as well as interpersonal relationships.

For its part, the adaptation of the Psychosocial Risk Assessment Questionnaire in Non-Formal Contexts (PRAQ-NFC) established three score intervals for the reference population. Each of these three intervals classified the study population into three exactly equal groups, establishing a traffic light around health. Thus, in the green interval were the scores with the most favorable psychosocial levels for health; on the contrary, the red interval established a more unfavorable psychosocial exposure grade for health; and the yellow interval defined the third of the employed reference population with intermediate psychosocial exposure levels for health [53].

The results obtained from the application of this questionnaire show that there were no significant differences according to sex in any of the dimensions analyzed. In relation to the qualification of the coaches, there were differences in the dimension “Psychosocial demands”, with the majority of level 2 coaches being in the risk zone, grade red, establishing relevance of the cognitive and emotional psychosocial demands: workload, time pressure and attention level that may be increased in pandemic times. In this sense, by worrying about the performance of their athletes and the completion of objectives, coaches perceive their functions as stressful and exhausting [28,29], especially in the conditions of confinement where physical, technical, tactical and psychological practice may have been affected as the restrictions managed to be negative for all these factors due to fear and preoccupation with the contagion.

Similarly, they highlighted the difficulty of balancing their work and their family, the support they received from their colleagues and superiors, and the quality of leadership governed by the clarity and role conflict in which they may find themselves. In this line, greater stress is usually found in professionals who have direct contact with people [25], such as coaches who are characterized by good performance, greater commitment to their work and high expectations in the achievement of objectives. However, in the COVID-19 time, this direct contact has diminished significantly, which may have a negative impact on their work.

In relation to the years of experience variable, the data indicate differences in the dimension “Psychosocial demands”, with coaches who have between 6 and 10 years of experience being those who are in the risk zone, yellow grade. In the dimension “Active work and development possibilities”, both those with more than 11 years of experience and the group with between 6 and 10 years of experience were in the yellow zone. This result can be explained by the working conditions currently experienced by the coaches. It is necessary to delve in these stress symptoms since they are not observed from the first moments when the coach enters the risk zone, being able to trigger the burnout syndrome [21].

There is a directly proportional relationship between stress and the dimensions “Psychosocial aspects” and “Insecurity”, the continuity of the coach in his job favors the formative processes and thus emotional stability [64]. This relationship between stress levels and “insecurity” may be a consequence of the pandemic because COVID-19 has caused some economic uncertainty and possible job and emotional instability that may increase stress levels in trainers, affecting the psychosocial aspect.

In turn, the data show the existence of an inversely proportional relationship between stress and the dimensions “Active work and development possibilities” and “Social support and leadership quality”. Coaches who scored higher in both dimensions showed low scores in psychosocial distress. A coach has to develop competencies and interpersonal skills to interact effectively in the professional and personal sphere, as well as conflict resolution skills and mastery of critical situations [30]. Such situations are sometimes caused both by the lack of experience and training and by the proximity of age between young coaches and players in junior (youth) and senior categories, making the management of interpersonal relationships in the group more complicated, not being as fluid as they should be between the figure of the coach and the player [26,27]. In addition, in coaches with more years of experience, COVID-19 did not influence their personal and work situation.

It is observed that coaches are constantly exposed to psychological discomfort and psychosocial risks because their work is performed with other people with different roles, including managers, coordinators, coaches, athletes, family and the public, among others; they assume situations of pressure, stress and insecurity, have a high cognitive and emotional load, and the active work and the possibilities of development depend on a higher degree of experience, through practical knowledge, to suffer less stress and psychosocial risks.

Regarding the limitations of the research, it is necessary to continue improving the instruments, as well as to apply new scales that measure the anxiety state factors, specific traits for coaches and resilience in this population under study, and it is necessary to adapt them to guarantee their reliability and internal validity to this population, as well as to expand the sample to know the reality that coaches live since there are few investigations that focus on coaches. Likewise, it is necessary to carry out a longitudinal study in order to follow up on and be able to generalize the results. In addition, it is necessary to continue researching and expanding the research environment itself and to expand it to other sports specialties in order to draw profiles according to stress and psychosocial factors since the work was carried out in basketball specialty.

This work opens the door to future research with coaches of all specialties and not only work with players, establishing the need for sports clubs to implement processes of prevention, regular monitoring and evaluation of psychological distress and psychosocial risks, to ensure the health of their coaches. On the other hand, it is necessary to continue researching and expanding the research environment itself and to carry it out in other sports specialties in order to draw profiles according to stress and psychosocial factors since the work was carried out in basketball specialty.

## 5. Conclusions

After using the adapted Psychological Distress scale and the Psychosocial Risk Assessment Questionnaire in Non-Formal Contexts with basketball coaches in the province of Granada (Spain) in the development of their sports work during the current 2020–2021 season characterized by the COVID-19 pandemic, we conclude the following:

In terms of the Psychological Distress scale, the scores were related to the age of the coaches and their years of experience, with the youngest and those with less experience showing the highest levels of stress. There were no differences according to the sex variable, although the low percentage of female basketball coaches in relation to the percentage of male coaches was observed.

In terms of basketball qualifications, level 1 monitors and level 2 coaches were those with the highest levels of stress. Level 3 top coaches were the ones who showed low levels of stress.

It can be concluded that there is a direct relationship between stress and the dimensions “Psychosocial aspects” and “Insecurity”: the higher the stress levels, the higher the scores in psychosocial aspects and insecurity.

An inversely proportional relationship was observed between stress and the dimensions “Active work and development possibilities” and “Social support and leadership quality”, so the coaches who scored high in both dimensions showed low scores in stress.

Regarding the PRAQ-NFC instrument, it was concluded that there were no significant differences in the sex variable in any of the dimensions analyzed.

With the PRAQ-NFC, the profile was established according to the basketball qualification in order to know the zone in which the basketball coaches were located. The data show that the top coaches (level 3) were in the green zone for the dimension “Active work and development possibilities” and in the yellow zone for the dimension “Social support and leadership quality”. It was the level 2 coaches followed by the level 1 monitors who were in the risk red zone, although it is necessary to complete these results with qualitative studies to know their situation and prevent future risks of psychosocial distress.

In terms of the years of experience variable, the data indicate differences in the dimension “Psychosocial demands”, with the coaches with 6–10 years of experience being those who were in the risk zone, yellow level. In the dimension “Active work and development possibilities”, both those with more than 11 years of experience and the group with 6–10 years of experience were in the yellow zone.

There was an inversely proportional relationship between stress and the dimensions “Active work and development possibilities” and “Social support and leadership quality”; the coaches who scored high in both dimensions showed low stress scores.

Finally, in terms of the levels of psychological discomfort, the participants with high levels of stress were in the red zone for the “Esteem” dimension and in the yellow zone for the rest of the dimensions; therefore, it is necessary to work with this group of coaches to avoid psychological discomfort and mental wear and tear.

## 6. Patents

There are no patents resulting from the work reported in this manuscript.

## Figures and Tables

**Table 1 ijerph-18-08722-t001:** Participants according to years of experience and basketball qualifications.

		Years of Experience				Total
		*N* %	1–5 Years	6–10 Years	11–25 Years	
**Basketball qualifications**	Monitor (level 1)	*N*	35	3	4	42
		%	38.5%	3.3%	4.4%	46.2%
	Coach (level 2)	*N*	6	17	11	34
		%	6.6%	18.7%	12.1%	37.4%
	Top coach (level 3)	N	0	1	14	15
		%	0.0%	1.1%	15.4%	16.5%
**Total**		N	41	21	29	91
		%	45.1%	23.1%	31.9%	100%

**Table 2 ijerph-18-08722-t002:** Descriptive analyses and reliability of the Psychological Distress in Non-Formal Contexts scale.

In the Past Month, How Often Did You Feel…	Media	SD *	Asymmetry	Total Correlation of the Corrected Elements
1. Tired for no good reason.	1.86	0.712	0.390	0.676
2. Nervous.	1.88	0.760	0.351	0.682
3. So nervous that nothing could calm you down.	1.11	0.343	3.37	0.652
4. Desperate.	1.43	0.680	1.745	0.650
5. Restless or uneasy.	1.78	0.642	0.487	0.484
6. So impatient that you could not keep still.	1.45	0.666	1.427	0.665
7. Depressed.	1.45	0.697	1.651	0.734
8. That everything you do represents a great effort.	1.68	0.763	0.752	0.593
9. So sad that nothing could animate you.	1.26	0.438	1.141	0.340
10. Useless.	1.27	0.552	1.998	0.639

* SD = Standard Deviation.

**Table 3 ijerph-18-08722-t003:** Correlation between the stress level and years of experience and basketball qualification.

		Stress Level
Basketball qualification	Pearson correlation	−0.254 *
	Sig. (bilateral)	0.013
Years of experience	Pearson correlation	−0.344 **
	Sig. (bilateral)	0.001

* Correlation is significant at the 0.05 level (bilateral). ** Correlation is significant at the 0.01 level (bilateral).

**Table 4 ijerph-18-08722-t004:** Descriptives of the Psychosocial Risk Assessment Questionnaire in Non-Formal Contexts.

Dimensions	Media	SD *	Asymmetry	Minimum	Maximum	Cronbach’s Alpha
1 Psychosocial demands	11.01	2.28	0.358	6.00	17.00	0.571
2 Active work and development opportunities	32.27	5.13	−0.162	20.00	40.00	0.842
3 Insecurity	6.54	2.49	0.953	4.00	13.00	0.738
4 Social support and leadership quality	30.65	4.44	−0.017	20.00	37.00	0.853
5 Double presence	6.05	2.51	1.05	3	14.00	0.422
6 Esteem	13.03	2.19	−0.686	6.00	16.00	0.800

* SD = Standard deviation.

**Table 5 ijerph-18-08722-t005:** Levels of the sections that make up the Psychosocial Risk Assessment Questionnaire in Non-Formal Contexts according to risk area in points.

Dimensions	Green	Yellow	Red
1 Psychosocial demands	From 6 to 10	From 11 to 12	From 13 to 17
2 Active work and development opportunities	From 35 to 40	From 29 to 34	From 20 to 28
3 Insecurity	From 4 to 5	From 6 to 7	From 8 to 13
4 Social support and leadership quality	From 35 to 37	From 28 to 34	From 20 to 27
5 Double presence	From 3 to 4	From 5 to 6	From 7 to 14
6 Esteem	From 15 to 16	From 13 to 14	From 6 to 12

**Table 6 ijerph-18-08722-t006:** Relationship between the components of the Psychosocial Risk Assessment Questionnaire in Non-Formal Settings and Stress Levels.

Dimensions	Stress Levels Average Range $\bar{x}$	Kruskal-Wallis *H*	*p*
1 Psychosocial demands	Low stress level = 46.82 $\bar{x}$ = 10.77	12.059	0.002
	Moderate stress level = 38.65 $\bar{x}$ = 10.28		
	High stress level = 62.75 $\bar{x}$ = 12.55		
2 Active work and development opportunities	Low stress level = 57.94 $\bar{x}$ = 34.25	7.099	0.029
	Moderate stress level = 43.73 $\bar{x}$ = 31.58		
	High stress level = 40.15 $\bar{x}$ = 30.83		
3 Insecurity	Low stress level = 35.48 $\bar{x}$= 5.48	10.498	0.005
	Moderate stress level = 50.56 $\bar{x}$ = 6.84		
	High stress level = 58.04 $\bar{x}$ = 7.41		
4 Social support and leadership quality	Low stress level = 59.11 $\bar{x}$ = 32.12	9.910	0.007
	Moderate stress level = 44.86 $\bar{x}$ = 30.25		
	High stress level = 36.79 $\bar{x}$ = 29.41		
5 Double presence	Low stress level = 44.90 $\bar{x}$= 5.74	2.578	0.247
	Moderate stress level = 44.91 $\bar{x}$ = 5.89		
	High stress level = 55.06 $\bar{x}$ = 6.70		
6 Esteem	Low stress level = 58.15 $\bar{x}$= 13.74	7.504	0.023
	Moderate stress level = 41.41 $\bar{x}$ = 12.64		
	High stress level = 43.65 $\bar{x}$ = 12.75		

## Data Availability

The datasets generated and/or analyzed during the current study are not publicly available, but are available from the corresponding author upon reasonable request.

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
