# Peer review of "Psychological Distress and Psychosocial Factors in the Non-Formal Context of Basketball Coaches in Times of the COVID-19 Pandemic"

_ijerph, 2021, doi:10.3390/ijerph18168722_

Round 1

Reviewer 1 Report

This manuscript addresses psychological distress in a non-probability sample of basketball coaches in Spain. The title is a bit deceptive. since the instruments to collect data do not include items associating the current mental status to COVID-19, other than that it was conducted during the COVID-19 pandemic. The introduction addresses COVID-19 as additional psychosocial factors. However, it does not associate the findings to being exposed, at least from what I could understand. The methods are appropriate and can be replicated; results are logical and aligned with the methodology. Results are consistent with previous literature, thus there is no apparent difference between this population and other worker populations. I perceive an enormous effort from the authors to explain potential differences with other populations that might not be there. Furthermore, I did not read about any limitations to the study. One that immediately caught my attention is if the size of the "Top Coaches" group was comparable to the others, especially when the size for the inexperienced was zero.  

The manuscript needs a comprehensive grammar check. I could not clearly understand the sentence in rows 43-44. Plus other run-in sentences. Authors probably need to translate the word "De" in table 5, or specify what "De" means.

Author Response

Dear Reviewer,

Thanks for considering paper entitled ‘Psychological distress and psychosocial factors in the non-formal context of basketball coaches in times of pandemic by COVID-19’ with reference ijerph-1309921, for possible publication in IJERPH. We would also like to express our gratitude for the valuable comments and suggestions made for the reviewers, which have helped us to improve the original paper significantly. We have considered the suggestions given by the reviewers and the original version has been mainly modified as follows:

Reviewer 2 Report

Manuscript Number: IJERPH-1309921

Title: Psychological distress and psychosocial factors in the non-formal context of basketball coaches in times of pandemic by COVID-19

Thank you for your interesting report. This report presented “Psychological distress and psychosocial factors in the non-formal context of basketball coaches in times of pandemic by COVID-19.” However, I do not think this paper is worth publishing. The reasons are as follows.

  1. You mentioned about ex-post-facto study with a single group retrospective design. I could not find what this design is, what are the strength of this design, and why it was chosen in this study.
  2. You mentioned about Kessler’s Psychological Distress Scale. I could not find how it was developed by whom, what language it was originally developed in, whether there is a validation study according to the Spanish translation, how many points are total, and what is the cutoff value.
  3. The title of this study has COVID-19, but there is no analysis related to COVID-19. It doesn’t appear to be related to COVID-19, except that this study was conducted in 2020. For example, there is no change pattern before or after COVID-19.
  4. The purpose of the study should be stated in the introduction part.
  5. In Table 2, writing the full names of DT is needed.

Author Response

(The authors gave the same response as above.)

Reviewer 3 Report

Dear Authors Congratulations on your research. In the attached file you can see my comments 

Author Response

(The authors gave the same response as above.)

Round 2

Reviewer 2 Report

Thank you for your effort.